# Towards Verifiable Declarative Mappings: A Vision Paper

Eduard Kamburjan[1,2], Mário Pereira[3]

[1]*IT University of Copenhagen, Denmark*
[2]*University of Oslo, Norway*
[3]*NOVA School of Science and Technology & NOVA LINCS, Lisbon, Portugal*

### Abstract
Declarative mapping languages such as RML improve the quality of knowledge graph construction by providing an abstraction layer through a domain-specific language. However, they do not *formally guarantee* correctness of the resulting knowledge graph, e.g., w.r.t. graph shapes: it is not guaranteed that every constructed graph will adhere to a set of given shapes. In this paper, we present our vision of validated-by-construction knowledge graphs: By giving declarative mapping languages a *verifiable semantics* in terms of a functional language, we are able to use program logics to verify whether the mapping ensures correctness. We present an example of a verified RML mapping and sketch further work on function and verifiable semantics for RML.

### Keywords
Declarative Mappings, Program Verification, Functional Languages

## 1. Introduction

Generating correct graph data from non-graph sources is a difficult task, which has led to the development of specialized, declarative languages such as RML [1]. While such languages simplify development and, thus, hopefully result in higher quality mappings, they by no means guarantee the correctness of the constructed knowledge graph: There is no tool to analyse whether a declarative mapping *always* generates a knowledge graph that satisfies certain SHACL shapes. Consequently, a time-consuming validation step has to be applied after generation, and bugs in the knowledge graph construction are caught only late in the development process.

Indeed, a formal semantics, which is a requirement for such an analysis, for the declarative language RML has only been proposed recently [2, 3]. It is designed in an algebraic fashion and aims to provide formal arguments for optimizations, akin to semantics of relational database queries. Formal semantics for correctness of programs, however, relies on denotational, axiomatic or operational semantics. Thus, carrying over techniques from program verification and analysis is not directly possible.

We report on our vision and on-going work for a verifiable semantics for RML. The core idea is to map RML into OCaml and have an OCaml function expressing the semantics of the mapping. While such a function is executable, the main goal is to use verification tools for OCaml, in particular Cameleer [4] and its specification language Gospel [5]. Thus, the semantics can focus on being *easy to verify*, not easy to optimize. In our semantics, the task to check whether an RML mapping ensures that its output satisfies a SHACL shape reduces to check whether the translation of the shape into a FOL equivalent [6] is a postcondition for the semantics of the mapping as a function. This is illustrated in fig. 1.

In the following, we give mapping and shapes in section 2, give its the functional semantics in section 3, where we express the mappings as an OCaml function and verify this function in section 4.

## 2. Motivating Example

We use an example to illustrate functional and verifiable semantics, taken and further simplified from prior work on dependency analysis of knowledge graph construction pipelines [7]. We note

---

*KGCW'26: Workshop on Knowledge Graph Construction*

✉ eduard.kamburjan@itu.dk (E. Kamburjan); mjp.pereira@fct.unl.pt (M. Pereira)

🄳 0000-0002-0996-2543 (E. Kamburjan); 0000-0003-4234-5376 (M. Pereira)

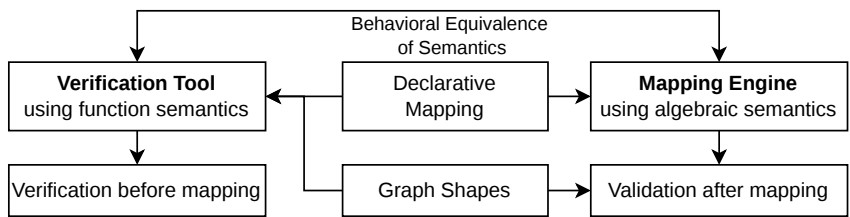

**Figure 1:** Overview over our vision: Early static verification (left) over post-hoc validation (right). The two semantics are linked on a theoretical level, but target different application scenarios and implementations.

that the referred to static analysis has no soundness proof – and one of the reasons was the lack of a formal semantics for RML at that point. Similarly, prior work on deriving SHACL shapes from RML [8] is not formally proven correct. The example, including its semantics, is available online under https://github.com/mariojppereira/rml_verify.

**Example 1.** *Figure 2 gives two RML maps in YARRRML syntax [9]. The first map `roles` iterates over a CSV file containing the id, names, and role of users, and creates a new role per row. Roles are uniques, as the same URI is used if a role occurs more than once. The second map `persons` iterates over the same, but creates one user per row. It links to the existing roles in its join condition, where the object for the created triple is taken from the subject map of the first map `roles`, if the field `role` in the source of both maps is equal. The SHACL shapes specifies that each user has exactly one role.*

We address the following question: How can we formally prove that every CSV file that is mapped by the above two mappings results in a knowledge graph validated by the given SHACL shape? It quite easy to see that this should be the case, but a formal proof requires (a) a formal semantics for the mapping, (b) a formal semantics for the correctness condition, (c) a proof system that verifies a given mapping against a given correctness condition. For RML and SHACL, requirement (b) is given: SHACL does have a formal semantics that can be exploited. There is no proof system for condition (c), and we argue that the currently available semantics (a) are not easily suitable to design one. There is ample work on program verification, but to the best of our knowledge there is no system for algebraic semantics of the kind used for RML [2, 3].

YARRRML

```
1 roles:
2   sources: [{access: 'users.csv',
    referenceFormulation: csv}]
3   s: dp:$(role)
4   po:
5     - [a, dp:Role]
6     - [dp:roleName, $(role)]
```

SHACL

```
1 //Each user has exactly one role
2 dp:UserShape
3     a sh:NodeShape ;
4     sh:targetClass dp:User ;
5     sh:property [
6         sh:path dp:hasRole ;
7         sh:maxCount 1 ;
8         sh:minCount 1 ;
9     ] .
```

YARRRML

```
1 persons:
2   sources: [{access: 'users.csv',
    referenceFormulation: csv}]
3   s: dp:$(id)
4   po:
5     - [a, dp:User]
6     - [dp:name, $(first) $(last)]
7     - p: dp:hasRole
8       o:
9         - mapping: roles
10          condition:
11            function: equal
12            parameters:
13              - [pr1,$(role),s]
14              - [pr2,$(role),o]
```

```
1 id,first,last,role
2 1,Peter,Schmitt,Admin
```

**Figure 2:** The top two code blocks show RML mappings with a join between them. The bottom left is a SHACL shape specifying the RML maps, and the bottom right is an example CSV file. Taken from [7].

```ocaml
OCAML
1  type person = {  nid: string;  first: string;  last: string;  role: string;  }
2  let generate_role_quadruples persons =
3   persons
4   |> List.map (fun person -> (person, "https:/dep/ex\#" ^ person.role))
5   |> List.map (fun (person, subject) ->
6  [{subject; predicate = "rdf:type"; obj = "dp:Role"; map = ("ROLES", person)};
7   {subject; predicate = "dp:hasName"; obj = person.role; map = ("ROLES", person)};])
8   |> List.flatten
```

**Figure 3:** OCaml implementation of `roles`, using two `map` calls for the subject and the predicate-object maps.

## 3. Towards Functional Semantics for RML

A declarative mapping is essentially a functional computation, i.e., functional program. Our core idea is to give a functional semantics to RML, i.e., a semantics as a function in a functional programming language, from semi-structured data to knowledge graphs. This semantics can be optimized towards *verification*: For execution, one has to prove that the functional semantics are equivalent to the algebraic ones (which are targeting mapping plans and are more suited for optimization). This proof of semantics equivalence has to be done only once, afterwards every mapping can be proven correct in isolation.

A RML mapping consists of a set of triples maps, which are applied per row (or similar) of the input data. Each triples map in turn consists of a subject map and a set of predicate-object maps. A triples map also has a logical source which is the source of the data. In the following, we only consider CSV.

The functional semantics of a triples map is a sequence of functional mapping operations. The first mapping operation is from a CSV row to a pair of (a) the subject and (b) the row the subject originates from. Every following mapping operation is a map from such pairs to quintuples, defined by a predicate-object map. The quintuples consist of an RDF triple, the name of the map and the input row. The later two elements are needed later for joining.

**Example 2.** *The code in fig. 3 shows the input type for the CSV logical source and the* `roles` *map. The map itself consists of two functional map functions. The first realizes the subject map, the second the two predicate-object map. Note that we encode an implicit precondition here: The CSV file has no missing value.*

Each map also has access to the quadruples resulting from prior maps. The order in which maps are run and chained, is defined by dependencies, where a map depends on another if it explicitly refers to it in a join. If a triples map contains a join, then the join condition becomes part of the defined mapping operator via a simple lookup on the quadruples defined by the prior map.

**Example 3.** *The code in fig. 4 shows the OCaml code for the* `persons` *map. The map itself consists again of two functional map functions, and one concatenation. The subject map and the first two predicate-object map are realized as before. The last map, which is based on a join involves retrieving all triples from the* `roles` *map, where the join condition holds on the input, and then copying over the subject into the object.*

The look-up for the join, which carries the input row, is not efficient. However, it spells out the join condition in a very explicit way that can be exploited by the proof – as discussed, the semantics does not aim for efficiency, but for clarity. Finally, the overall mapping also contains an additional step that maps the quadruples to RDF triples.

## 4. Towards Verifiable Semantics for RML

We use the *auto-active* deductive verification tool Cameleer [4], which is built on top of two main components: (i) a front-end, where the user supplies logically specified OCaml code, and (ii) a back-end,

```ocaml
                          OCAML
 1  let generate_person_with_roles_quadruples persons role_quadruples =
 2   persons
 3   |> List.map (fun person -> (person, mk_subj2 person)) (* subject map *)
 4   |> List.map (fun (person, subject) ->
 5    let basic_quads = [ (* first two predicate–object maps *)
 6    { subject; predicate = "rdf:type"; obj = "dp:person"; map = ("PERSONS", person) };
 7    { subject; predicate = "dp:hasName"; obj = person.first ^ " " ^ person.last;
 8      map = ("PERSONS", person) };
 9    ] in
10    let role_quads = (* last predicate–object maps *)
11     let matching_role_quads = List.filter (fun (quad : quadruple) ->
12      let (mapping_name, source_person) = quad.map in
13       source_person.role = person.role (* join conditions *)
14     ) role_quadruples in
15     List.map (fun (role_quad : quadruple) ->
16      { subject; predicate = "dp:hasRole"; obj = role_quad.subject; (* join *)
17        map = ("PERSONS", person) }
18     ) matching_role_quads
19     in
20     basic_quads @ role_quads (* concatenation *)
21    )
22   |> List.flatten
```

**Figure 4:** OCaml implementation of persons.

that translates the input program into so-called *Verification Conditions* (VCs), i.e., logical formulae that capture the formal correspondence between code and specification. Cameleer is *mostly automatic*: instead of relying on heavy-manual user interaction, the tool uses SMT solvers to discharge the generated VCs to automatically find a proof for the given formula. OCaml code is annotated using Gospel [10], the *G*eneric *O*caml *SPE*cification *L*anguage, which integrates OCaml constructs into first-order specification, and we envision the generation of such specification from SHACL.

Gospel specifies functions in terms of *contracts*: on one hand, the *pre-condition*, i.e., a logical description of the state under which the function is supposed to be called; on the other hand, the *post-condition*, i.e., a formula that captures the final state after the function executes, if it terminates and the pre-condition is initially met. Consider the following piece of OCaml-annotated code:

```ocaml
                          OCAML
 1  let admin_role (ps: person list) : person list =
 2    List.map (fun p -> {p with role = "admin"}) l
 3  (*@ r = translate ps
 4      ensures forall e. List.mem e r -> e.role = "admin" *)
```

It features a simple translation function, from list of persons into a new persons values. We use here the type person defined in Fig. 3. The admin_role function above produces a new list where every person role is set to the string "admin". The Gospel contract is given after function definition. The first lines introduce a name for the result value, here r. The next line introduces a post-condition, via the ensures clause. It states a *universal* property on all the elements of the result list, introduced by the forall quantifier. This kind of list transformation functions, implemented on top of pure, higher-order functions, are pervasive and idiomatic in the functional programming paradigm. Our goal is to build our RML verifiable semantics pipeline on top of such features, leveraging on functional programming key features and associated verification tools to automatically reason about RML semantics.

```ocaml
OCAML

1 let generate_person_with_roles_quadruples persons role_quadruples =
2   ... (* see figure 3 *)
3 (*@ result = generate_person_with_roles_quadruples persons role_quads
4     ensures exists r h. List.mem r result /\ List.mem h r /\
5         r.subject = h.subject /\ r.predicate = "rdf:type" /\ r.obj = "dp:User" *)
6
7 let quadruples_to_triples (quadruples : quadruple list) : triple list =
8   let triples = List.map quadruple_to_triple quadruples in
9   remove_duplicate_triples triples
10 (*@ result = quadruples_to_triples quadruples
11     ensures exists h r. r. List.mem r result /\ List.mem h result /\
12         r.subject_triple = h.subject_triple /\ r.predicate_triple = "dp:type" /\
13         r.obj_triple = "dp:User" *)
```

**Figure 5:** Partial example specification: Each generated node with type `dp:Role` is the subject of two triples.

**Verification of translated SHACL shapes.** We propose to apply the Cameleer-Gospel verification methodology to our motivating example of section 2. The specification of the main translation functions is given in Figure 5. We consider the implementation of function `generate_person_with_roles_quadruples` as presented in Figure 3.

The post-condition of function `generate_person_with_roles_quadruples` states that there *exist* two elements r and h in the transformed list, such that r and h have the same `subject` string. It also states the fields `predicate` and `obj` of element r are, respectively, equal to strings `"rdf:type"` and `"dp:User"`. Contrary to a universal specification, a proof of an existential property amounts to search for a *witness*, i.e., a value that indeed respects the given formula. Finally, the post-condition of function `quadruples_to_triples` follows the same pattern, the only relevant difference being the conversion between values of type `quadruple` to `triple`. For brevity's sake, this is only a partial proof and specification of the SHACL shape, namely that a triple exists, which we can check the property path against. The full specification, which validates that there is exactly one triple with predicate `dp:hasRole` is in the auxiliary material. Cameleer, using Why3 [11] as its backend, is able to *fully automatically* prove Figure 5: All of the generated VCs are automatically discharged by SMT solvers.

Currently, the functional correctness of the RML semantics is hand-written as a Gospel post-condition. Our vision is, however, to build a verification pipeline where the correctness property should be given directly at the level of the SHACL shapes and then automatically translated into an OCaml program.

## 5. Conclusion

**Related Work.** Our approach is related to other lines of work on correctness in the semantic web, such as the graph updates of Ahmetaj et al. [12]. Graph updates are formal updates of knowledge graph that can be verified to preserve validity of SHACL constraints. They defined mappings between two graphs, not from semi-structured data into a graph. Regarding verification, the integration of Description Logics as a specification language into a program logic has been recently investigated [13]. As an orthogonal approach, Ileri and McGinty [14] formally verified a Description Logic reasoner, addressing not mappings, but the software tools underlying the semantic web.

**Vision.** Our long-term goal is a tool that can remove the need to run SHACL validation during knowledge graph construction and evolution, because each used step is verified to establish, or preserve, correctness *by construction*. In this work, we sketch how a formal foundation for such a tool could look like, and how it relates to the formal semantics that form the basis of the actual mapping engine.

We see it as an advantage that our vision does not rely on the implementation or design of new languages or program logics. By using established technologies for the mapping (RML), its semantics

(OCaml), its specification (SHACL) and the proof system (Cameleer), we avoid many of the problems that plague deductive verification [15], in particular the notorious specification bottleneck [16].

While we only give one example, it is already structured according to the RML metamodel, and we take it as a first indication that the approach can be systematized. In particular, we conjecture that the auxiliary specification between the mappings can be partially generated as well, thus reducing the need for human written OCaml-specification. The next step is to systematically define the functional semantics of full RML and prove its equivalence with the algebraic semantics.

## Declaration on Generative AI

The authors used Claude code in order to: Prototype the initial OCaml code. After using this tool, the authors reviewed and edited it and take full responsibility for the publication's content.

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
