# OpenReview forum: "Towards Verifiable Declarative Mappings: A Vision Paper"
_eswc-conferences.org/ESWC/2026/Workshop/KGCW — KGCW 2026_

### Official Review · ~Ioannis_Dasoulas1 · 2026-04-03
**Interesting vision paper for knowledge graph validation during construction**

**Rating:** 7
**Confidence:** 3

**Review:**

This vision paper proposes an approach for validating knowledge graph construction during construction and not as a separate step. The method maps RML rules to OCaml in a formal manner to combine construction and validation.

Strengths:
1. Relevant problem: The authors tackle an important problem of knowledge graph construction pipelines, the generation of wrong triples which can only be identified after full materialization. As graphs scale, this problem becomes more apparent.
2. Useful example: The authors provide a useful example for their approach.
3. Validation ordering: The hierarchical validation that the authors propose through dependency chains can potentially have great benefits for saving up time in knowledge graph construction validation.

Weaknesses:
1. Lack of Graph Maps: The method does not consider graph maps (minor comment given that it is a vision paper)
2. Formalizations: The authors correctly mention the lack of formal notations in the field, until recently, but proceed to also not use any formal notations. I understand that this is a vision paper but at least some basic formal notations would be helpful
3. Motivation could be strengthened: The authors mention that validation after full generation of triples is time-consuming but there is no indication whether their method should be less time-consuming. The motivation could be strengthened by focusing on the advantages of a formal verification method during knowledge graph construction, such the benefit of having a standard method and hierarchical validation.


Minor comments:
1. ‘Uniques’ should be ‘unique’ in Example 1 description
2. ‘It quite easy’ should be ‘It is quite easy’ in Section 2

---

### Official Review · ~Ana_Iglesias-Molina2 · 2026-04-03
**Nice contribution for an interesting line of research, missing a bit of broader vision**

**Rating:** 7
**Confidence:** 4

**Review:**

This vision paper reflects on the idea of validating KGs prior to their construction leveraging declarative mapping languages, instead of running the validation after the construction. The authors propose to combine mapping rules and SHACL shapes in OCaml to this end, to be able to verify that the shapes are fulfilled in the mapping rules. The paper shows clear examples of how this can be achieved.

In general I liked the idea and purpose of the paper, being able to "skip" the validation step saves time and another step in a KG lifecycle. I miss however two reflections that I consider essential to really make possible the authors' vision:

What if the restriction can only be seen in the data? The example in the paper is perfect for this: the mapping specifies that each person can have one role, and for this there is only one mapping rule. But without also checking the data, it is possible that this constraint is violated if given a CSV like this:

Person, ..., Role
Person1, ..., Role1
Person1, ..., Role2
Person2, ..., Role1

I agree that being able to perform some pre-validation is achievable with mappings, but maybe not everything. I'd like to read the author's opinion on this issue.

The other issue is regarding automation: IMO this approach can be adopted whenever it doesn't introduce new steps in the KG construction, i.e. automating the RML translation to OCaml and running this as a prior step integrated in a KGC engine. Otherwise, it'd be forcing the user to learn (yet another) language appart from SHACL to perform these validations. Ideally, the user would only need to write the mappings and shapes, then the engine takes care of the rest. Is this possible with the authors proposal? There are some sentences mentioning this but I'd like to read an extended vision on the architecture of how this would work.

Having said that, I see this paper as an interesting line of research, and a valuable contribution for the workshop for sparking the discussion. I hope more space is allowed to add some reflections about the raised issues.

Minor:
- OCaml is intensively mentioned in the paper but not explained, a brief one-line description in the introduction where it's first mentioned would be helpful. Similarly, there are other acronyms not explained throughout the

---

### Official Review · ~Maria-Esther_Vidal2 · 2026-04-04
**Vision paper proposing a paradigm for “validated-by-construction” knowledge graphs**

**Rating:** 9
**Confidence:** 5

**Review:**

Summary:
This vision paper proposes a paradigm for “validated-by-construction” knowledge graphs, where correctness is guaranteed during the mapping process rather than verified post hoc. The core idea is to reinterpret RML mappings as functional programs and provide them with a functional semantics (in OCaml) that enables formal verification using program logics.

By combining RML (mapping language), OCaml (semantic foundation), SHACL (constraint specification), and Cameleer/Gospel (verification tools), the authors aim to prove that mappings always generate graphs that satisfy the given constraints. The approach relies on defining mapping semantics as compositional functional transformations (e.g., from rows to RDF quadruples) and expressing correctness as postconditions derived from SHACL shapes, which can then be automatically verified using SMT-based reasoning.
The long-term vision is to eliminate runtime validation by ensuring correctness by construction, supported by a formally verified semantics and automated translation from SHACL to verification conditions.

Evaluation:
This vision work is grounded in formal semantics, program verification, and existing Semantic Web standards. The use of a concrete running example (RML, SHACL, and OCaml) strengthens the credibility of the proposal. However, the work remains at an early stage, with partial proofs and limited empirical validation.
The paper is original in several aspects: a) Recasting declarative mappings as functional programs. b) Bridging knowledge graph construction with deductive program verification. c) Proposing correctness-by-construction for KGs, rather than validation-after-generation. d) Integrating RML + SHACL + formal verification tools in a unified pipeline. This represents a new research direction at the intersection of Semantic Web, data integration, and formal methods.
Positive Points (PPs)
-) PP1. Conceptual originality and vision. The paper introduces a novel perspective by framing declarative mappings as functional programs and enabling correctness-by-construction for knowledge graphs.
-) PP2. Strong theoretical foundation. The approach is well-grounded in formal semantics and program verification, leveraging established technologies (RML, SHACL, OCaml, Cameleer) in a consistent way.
-) PP3. High potential impact. If realized, the proposed framework could eliminate costly post-hoc validation and significantly improve the reliability and trustworthiness of KG construction pipelines.
Negative Points (PNs)
-) PN1. Limited maturity and validation. The work remains at a conceptual stage, with only partial proofs and no large-scale or real-world evaluation.
-) PN2. Scalability and practicality concerns. The feasibility of applying functional semantics and deductive verification to complex, large-scale mappings is not demonstrated.
-) PN3. Complexity and adoption barriers. The approach requires expertise in formal methods and functional programming, which may hinder adoption within the broader Semantic Web community.

Overall Assessment:
This is an original vision paper that opens a promising research direction by connecting knowledge graph construction with formal verification. While still preliminary, it provides a convincing conceptual foundation and a clear long-term agenda. My recommendation is for acceptance

---

### Decision · Program_Chairs · 2026-04-09

**Decision:**

Accept

**Comment:**

This paper has been selected for presentation at the KGC workshop. We strongly encourage the authors to consider the reviews whilst revising the paper. Camera-ready instructions will soon follow.